# Clinical Application of Adipose Derived Stem Cells for the Treatment of Aseptic Non-Unions: Current Stage and Future Perspectives—Systematic Review

**DOI:** 10.3390/ijms23063057

**Published:** 2022-03-11

**Authors:** Amarildo Smakaj, Domenico De Mauro, Giuseppe Rovere, Silvia Pietramala, Giulio Maccauro, Ornella Parolini, Wanda Lattanzi, Francesco Liuzza

**Affiliations:** 1Department of Aging, Neurological, Orthopaedic and Head-Neck Sciences, Fondazione Policlinico Universitario Agostino Gemelli IRCCS, 00168 Rome, Italy; amarildo.smakaj@gmail.com (A.S.); demaurodomenico@gmail.com (D.D.M.); rovere292@hotmail.com (G.R.); sy.pietramala@gmail.com (S.P.); giulio.maccauro@unicatt.it (G.M.); 2Department of Geriatrics and Orthopaedic Sciences, Università Cattolica del Sacro Cuore, 00168 Rome, Italy; 3Department of Life Science and Public Health, Università Cattolica del Sacro Cuore, 00168 Rome, Italy; ornella.parolini@unicatt.it; 4Fondazione Policlinico A. Gemelli IRCCS, 00168 Rome, Italy

**Keywords:** non-unions, pseudoarthrosis, adipose derived stem cells, ADSCs, mesenchymal stromal cells, MSCs, regenerative medicine, bone regeneration

## Abstract

Fracture non-union is a challenging orthopaedic issue and a socio-economic global burden. Several biological therapies have been introduced to improve traditional surgical approaches. Among these, the latest research has been focusing on adipose tissue as a powerful source of mesenchymal stromal cells, namely, adipose-derived stem cells (ADSCs). ADSC are commonly isolated from the stromal vascular fraction (SVF) of liposuctioned hypodermal adipose tissue, and their applications have been widely investigated in many fields, including non-union fractures among musculoskeletal disorders. This review aims at providing a comprehensive update of the literature on clinical application of ADSCs for the treatment of non-unions in humans. The study was performed according to the Preferred Reporting Items for Systematic Reviews and Meta-Analyses (PRISMA). Only three articles met our inclusion criteria, with a total of 12 cases analyzed for demographics and harvesting, potential manufacturing and implantation of ADSCs. The review of the literature suggests that adipose derived cell therapy can represent a promising alternative in bone regenerative medicine for the enhancement of non-unions and bone defects. The low number of manuscripts reporting ADSC-based therapies for long bone fracture healing suggests some critical issues that are discussed in this review. Nevertheless, further investigations on human ADSC therapies are needed to improve the knowledge on their translational potential and to possibly achieve a consensus on their use for such applications.

## 1. Introduction

Bone fractures are the most common traumatic injuries affecting large organs in humans. Approximately 5–10% of patients experience impaired fracture healing, and a subset of them will develop non-union [1], defined as the failure to achieve bony union by 9 months since the injury, for which no signs of healing has occurred for 3 months [2]. Fracture non-union is a painful chronic condition with great negative impact on the quality of life, significant medical treatment costs and delayed return to work [3,4,5]. The treatment of aseptic non-unions includes different surgical fixation strategies to improve the biomechanical properties of the fractured bone. Among these, autologous bone grafting and stable fixation are considered the gold standard. In fact, bone autografts do not only fill the gap, but they also enhance local biology. However, the limited yield of endogenous stem cells, with the secreted bioactive signals derived thereof, in autologous bone, especially in the elderly or in selected comorbidities, makes the autograft often fail to promote a functional and biologically effective bone healing. To address this criticality, biological therapies have been introduced, such as platelet-rich plasma (PRP), bone morphogenetic proteins (BMPs) and autologous mesenchymal stromal cells isolated from bone marrow (MSCs) [6,7]. The morbidity of bone marrow harvesting along with the low cell isolation yield from this site fostered the search for additional tissue sources for somatic stem cells. The latest research has been indeed focusing on adipose-derived stem cells (ADSCs) [8]. ADSC are multipotent cells which can be quite easily isolated from the stromal vascular fraction (SVF) of liposuctioned hypodermal adipose tissue [9,10]. Their applications in musculoskeletal disorders have been widely investigated, especially for cartilage and tendon regeneration [11,12,13]. As far as bone regeneration is concerned, many experimental approaches in preclinical models and diverse clinical trials tested the efficacy of autologous ADSC and/or SVF transplantation in promoting effective bone healing of critical-size cranial and craniofacial defects [14,15]. Moreover, there are several clinical trials testifying the regenerative potential of ADSCs in different fields: diabetes, Crohn’s disease, wound healing, liver cirrhosis and many other diseases [16,17,18]. However, the literature reporting ADSC applications for improving bone regeneration in non-union fractures is still poor and fragmentary, suggesting some critical aspects that hamper their clinical translation. The aim of the study is to provide an up-to-date review of the literature on the clinical application of adipose-derived stem cells for the treatment of non-unions in humans.

## 2. Materials and Methods

### 2.1. Study Design and Search Strategy

The present study is a systematic literature review reported according to the Preferred Reporting Items for Systematic Reviews and Meta-Analyses (PRISMA) guidelines [19] (Figure 1). MEDLINE via PubMed and Embase, Scopus, Cochrane Library database were searched using the keywords: “adipose derived stem cells”, “ASC”, “ADSC”, “adipose stem cells”, “non-union”, “bone healing”, “pseudoarthrosis” and their MeSH terms in any possible combinations using the logical operators “AND” and “OR”. The reference lists of relevant studies were screened to identify other studies of interest. The search was reiterated until 31 May 2021.

### 2.2. Inclusion and Exclusion Criteria

All studies published as full-text articles in indexed journals describing the application of adipose derived stem cells for the treatment of aseptic non-unions in humans were considered eligible. Only articles written in English were included. No date limits publication was established. Expert opinions, in vitro investigations, studies on animals, unpublished reports, abstracts from scientific meetings and book chapters were excluded from review.

### 2.3. Data Extraction and Analysis

Two observers (A.S. and S.P.) independently searched and collected data from the included studies. Any discordances were solved by consensus with a third author (F.L.). The following data were extracted by included studies: year of publication, types of research studies, demographic features of patients, diagnosis and symptoms, treatment performed, possible complications and outcomes and follow-up. All data concerning adipose stem cell harvesting, manipulation and application were carefully reviewed and collected. Numbers software (Apple Inc., Cupertino, CA, USA) was used to tabulate the obtained data. Categorical variables are presented as frequency and percentages. Continuous variables are presented as means and standard deviation. Only one decimal digit was reported and was rounded up.

## 3. Results

The electronic search of the literature consisted of 576 studies. After duplicate removal, 555 articles were screened for title and abstract, but only 134 papers were considered eligible, as all others were off topic. Two non-English written studies, 31 reviews and one book chapter were excluded. Ninety five studies were considered not eligible because they were performed on animal models and two studies were considered not eligible as they were performed in vitro. Therefore, full-text papers were carefully evaluated. Finally, three articles met our inclusion criteria [20,21,22]. The Level of Evidence (LoE) of the studies was V-IV. In fact, one paper included in the review was a case report, whereas two were case series. The latter cases were referring to the same experience of one center in two different moments of the trial [20,21]. Therefore, a total of 12 patients were included in the studies. Six patients had bone tumors. Among them, Three patients were affected by osteosarcoma, two patients were affected by Ewing sarcoma, and one patient was diagnosed with chondrosarcoma. In all the patients, tumor resection resulted in critical size-defects. Four cases were congenital non-unions affecting the right tibia and fibula in three cases and the left ulna in the last one. There were only two cases of acquired non-union affecting the sternum and the right tibia. Among the reported results, the mean age was 21 years and 11 months old, the male/female ratio was 3:1 and the mean follow-up period was 26 months and 25 days. Further demographic analyses were not possible due to the small sample of patients. No early acute side effects were reported. Patients were enrolled for advanced cell therapy after the failure of conventional therapies such as distraction osteogenesis with Ilizarov fixation, iliac crest autograft, and demineralized bone matrix alone. Fracture healing enhancement with SVF was performed as primary surgery in one case. Among the included articles, autologous stem cells harvesting was performed by lipoaspiration (Coleman’s procedure based devices). Two studies were performed using a scaffold-free osteogenic 3D adipose-derived graft (11 patients), whereas one study enhanced osteosintesis by ADCSs injection after completion of the plating (one patient). Adipose derived grafts were produced in line with good manufacturing practices (GMPs) and the ISO 9001–2008 quality management system. On the contrary, Khalpey et al. harvested SVF through minimal tissue manipulation (i.e., without enzymatic digestion, cell seeding and in vitro culture expansion), hence by-passing GMP-proof production, and enabling further effective grafting within the same surgical procedure. In case of stem cell manufacturing, the mean time necessary to obtain a three-dimensional osteogenic graft was 94.6 days.

As regards complications, two patients underwent bone graft and plate removal after Staphylococcus aureus and Enterococcus faecalis infection at more than 10 months post-transplantation. One patient had a subcutaneous collection due to Staphylococcus aureus, successfully treated without graft removal. No further complications were reported. Outcomes were described as general clinical conditions as none of the selected studies used clinical completed scores or patients reported outcomes measures (PROMs). During follow-up, all patients had good general clinical outcomes with fracture healing and bone defects filling, apart from three patients who revealed absence of consolidation demanding further surgery. The main demographical and clinical data of the included studies are shown in Table 1.

## 4. Discussion

### 4.1. Orthopaedic Overview on the Issue

Treatment of aseptic non-unions is a very challenging issue in the orthopaedic field. Giannoudis and co-workers in 2007 [23] described the so-called diamond shaped concept which introduces an additional element to the triangular biological model of bone healing: the mechanical environment of the fracture site. Therefore, the gold standard surgery aims to give sufficient stability and to provide osteogenesis, osteoinduction and osteoconduction by autologous bone grafting. Nonetheless, significant limitations of autograft include the limited availability of donor bone (usually from the iliac crest) along with the associated donor site morbidity (haematoma, seroma, dysaesthesia, paraesthesia, infection, vascular injury and iliac crest fracture). The development of the Reamer–Irrigator–Aspirator (RIA; Synthes, Paoli, PA, USA) enabled to overcome these limitations, providing a valuable tool to collect autologous bone with reduced morbidity and invasiveness. However, despite a number of research efforts, non-unions remain a social and economic burden. In fact, on one hand, psychological wellbeing of patients affected by tibial shaft non-union is worse than normal population. This can be easily assumed thinking about the repeated in-hospital stays and consecutive highly invasive surgical procedures required for the treatment of non-union, which predispose to the development of posttraumatic stress disorder syndrome (PTSD). On the other hand, long bone non-unions represent also a substantial economic burden. According to Tay et al. in 2014 [3], only 59% of patients suffering non-union are able to return to their job after twelve months, in comparison to the 72% of those who have no other complications after surgical intervention, with important economic implications.

### 4.2. Current Stage of ADSCs Human Application in Non-Unions

ADSCs are multipotent cells originated by mesoderm: they have the potential to differentiate towards adipogenic, osteogenic and chondrogenic lineages, in vitro [8]. Adipose tissue is a way more accessible source for multipotent stem cells being widely diffused among the body and containing extended stem cell depots. The adipose tissue stromal vascular fraction (SVF) comprises, besides ADSCs, a rich and heterogeneous cell population (preadipocytes, fibroblasts, vascular smooth muscle cells, endothelial cells, resident monocytes/macrophages, lymphocytes), plus a soluble fraction containing extracellular matrix components and the cellular secretome [10]. The latter includes several growth and endocrine factors with bone healing and angiogenic activities, which potentially act as a biological boost for functional fracture repair [24]. The average cellular yield in processed lipoaspirate is 2% of nucleated cells, with approximately 5000 fibroblast colony-forming units (CFU-F) per gram of adipose tissue [24,25], suggesting that it is an efficient source of multipotent stromal stem cells. The characterization of SVF stromal fraction and phenotyping of ADSC is the first pivotal step on the way to ADSC investigation. ADSCs meet most of the minimal criteria set by the International Society for Cellular Therapy (ISCT) to define human mesenchymal stromal cells (MSCs): in vitro trilineage (osteogenic, chondrogenic, and adipogenic), potential, plastic-adherence, expression of the MSC-specific antigens CD73, CD90 and CD105, and lack of hematopoietic lineage markers [24,26]. Bourin et al. in 2013 provided an initial guidance regarding the minimal properties expected for adipose tissue-derived cells, proposing some markers to phenotype SVF, ADSCs and to assess ADSC functionality [8]. Overall, these features have suggested this tissue source more attractive for regenerative medicine applications, compared with bone marrow, whose retrieval is inherently more invasive, painful, and often unsuccessful, and considering the lower yield in stem cell isolation. ADSCs and BM-MSCs exhibit virtually identical immunophenotype, in vitro trilineage potential, and transcription profiles, at least for genes related to the stem cell phenotype, supporting the concept of a common origin of the mesenchymal lineage from a wide variety of tissues [27]. Nonetheless, ADSCs can be maintained longer in culture and possess a higher proliferation capacity compared to BM-derived MSCs [27]. Also, they have been shown to be more genetically stable in long-term cultures [28] and displayed increased immunosuppressive properties [29].

A number of clinical devices are currently available for the rapid separation of adipose tissue fractions in closed systems, enabling the enzyme-free collection of a clinical grade ADSC-enriched SVF through minimal tissue manipulation for regenerative medicine applications in single surgical procedures [9].

Many studies in the literature investigated the efficacy of ADSC-based approaches for inducing bone regeneration and healing in in vitro and in vivo models. Animal models employed for bone tissue regeneration include critical-size calvarial defects, ectopic bone formation, long bone segmental defects, vertebral defects or fusions and mandibular defects [30,31].

However, according to our review of the literature, only three studies to date, have reported clinical outcomes after ADSCs application for bone healing in non-unions and bone defects, which represents an inherent limitation in this review. The limited clinical application of ADSCs in bone regeneration, despite the significant results obtained in preclinical studies, is mainly due to the limitations imposed by national regulatory frameworks. In fact, over the past two decades, Countries are facing the challenges of introducing advanced therapy medicinal products (ATMPs)—biological products with the deriving legislation and regulatory limitations—in the clinical practice. The European Union (EU) and the United States, (US) with their respective authorities, European Medicines Agency (EMA) and Food and Drug Administration (FDA), applied the concepts of “substantial manipulation” and “homo-functionality” for the classification and consequent regulation of ATMPs. For instance, cell separation, concentration, and purification do not represent a substantial manipulation whereas cell-culture expansion and activation with growth factors is a “more than minimally manipulation”. As far as homo-functionality is concerned, if the cells or tissues maintain their original function in the same anatomical or histological environment, they are meant to have “same essential functionality”. On the other hand, in case of non-homologous use, the cells or tissues are not intended to be used for the same essential function as in the donor site.

In this regard, the use of ADSC, SVF or other adipose tissue-derived bioactive fractions for treating bone defects inherently refers to “more than minimally manipulated” tissues for “non homologous use”. In the absence of substantial preclinical data from animal models supporting the efficacy of a given experimental therapy, the translation into clinical products is still hampered.

These considerations reflect the need to implement future scientific research in the field.

In the papers considered in this review, the most frequent indication was represented by bone defects resulting from wide bone tumor resections [20,21] while only one patient featured a post-traumatic non-union of long bones, which represents the most frequent cause of non-union [1,32]. Moreover, the majority of the population was younger than 18 years, which represents both a strength and a limitation of the studies. On one hand, data from pediatric case samples are rare and informative, as they are considered a special population, given the necessary ethical considerations regulating their involvement. On the other hand, non-union are more frequent in adults, where ADSC grafting could find a more suitable application in post-traumatic non-unions. Hence the patients considered in the papers reviewed here could not be completely representative of the non-union topic in the orthopedics trauma field.

Regarding the ADSCs application, two of the selected papers assessed the safety and feasibility of a scaffold-free osteogenic 3D adipose derived graft for the treatment of bone non-union [20,21]. They demonstrated the reproducibility of manufacturing of three-dimensional grafts based on autologous ADSCs. The procedures were not associated with serious side effects, adverse events or complications, and the results confirmed the safety and feasibility of ADSCs therapy. The studies concluded that scaffold-free adipose derived grafts safely lead to restoration of bone anatomy and function in nonunions, with minor donor site morbidity and no oncological side effects. However, scaffold-free grafts fail to exert biomechanical properties, needed for the weight bearing properties of selected skeletal sites (i.e., mostly in the appendicular skeleton), while they rather fill the bone defect and provide a biological burst to the site of non-union. Therefore, a proper fixation is mandatory after a scaffold-free adiposederived graft non-union site enhancement.

Khalpey et al. in 2015 [22] described the case report of a successful sternal reconstruction with autologous stem cells from the SVF. The patient had a chronic sternal nonunion with bone loss after coronary artery bypass grafting which, given the extensive bone defect, was treated with porous demineralized bone matrix and plating. At the end of the procedure, the surgical site was enhanced with SVF and lipoaspirate. The patient had a good outcome with nonunion healing at six months.

### 4.3. The Role of Bone Grafts

In aseptic nonunion, when there is a concomitant bone defect, grafts are decisive to achieve bone healing [33,34] as they provide osteogenesis, osteoinduction, and osteoconduction to the non-union site [35]. Bone grafts can be divided into natural and synthetic substitutes. Natural bone grafts, in turn, consist of autografts and allografts. Autografts are harvested from an anatomic site and transplanted to another site within the same individuals weather allografts are transplanted in different individuals of the same species. Synthetic grafts are bone substitutes which derive from a variety of material (e.g., calcium sulfate, calcium phosphate ceramics and cements) [36]. In recent years, their market experienced an important development, driven by the demand for new biocompatible materials with osteoconductive properties and an adequate availability to fill large bone defects [36,37]. Table 2 summarizes some of the most important characteristics of the above-mentioned bone grafts and substitutes [38].

Despite the presence of many alternatives, autograft is still considered the gold standard for the treatment of aseptic non unions regardless of the morbidity of the donor site [34]. In fact, it has the lowest immunogenicity and highest biocompatibility among the available bone grafts as well as lower costs.

Moreover, tissue engineering allows to seed ADSCs on biocompatible scaffolds or biomaterials to generate bone regenerative grafts [39]. In fact, Saçak et al. demonstrated that bioactive glass (S53P4) implanted with ASC is a biocompatible construct stimulating radiologically and histologically evident bone regeneration similar to autologous bone grafting in animal model critical size calvaria defects [40]. Furthermore, Mazzoni et al. demonstrated that in the hydroxylapatite-collagen hybrid scaffold, seeded ADSCs seems to be an excellent biomaterial able to drive bone regrowth and remodeling in vitro and in maxillofacial patients who underwent malar augmentation procedures [39]. At present, many scaffolds are emerging with potential applications in different fields of medicine, growing the interest toward tissue engineering-based grafts.

### 4.4. What’s New from Basic Science? Future Perspectives

Advanced cell therapy is more than a promise for orthopaedic surgery. Despite the still limited clinical reports, significant cues come from the numerous in vitro and in vivo preclinical studies, documenting the osteogenic properties of ADSCs and their interactions with different scaffolds, along with the search for osteo-inductive molecules, in the design of innovative tissue engineering strategies. Growth factors as FGF-2, TGF-β combined with ADSC on a polysaccharide-based tissue engineered periosteum have shown promising results as they promote a more stable harvesting and integration in murine models [41]. Furthermore, different studies proved the increased osteogenic properties of transgenic ADSCs overexpressing bone morphogenetic protein 2 (BMP2) [42] or BMP-7 [43] in promoting bone healing of long-bones defects. Nonetheless, despite the high transfection efficiency and yield in the transgene expression that they enable, the use of adenoviral vectors for gene transfer seems to complicate a rapid translation of such protocols in the clinical practice.

Innovative scaffolds are investigated as stable supports for ADSC harvesting. Some of the newest scaffolds include decellularized matrices as decellularized human adipose tissue (DAT) hydrogel [44], ceramics such as calcium phosphate-based cements, and bioglass [45] and synthetic polymers such as polyglycolic acid (PGA). Fibrin glue is also an option as a cell-delivery vehicle for ADSCs [46]. Also, an example on how adipose tissue itself can be used as a scaffold is Adiscaf, a construct obtained by human liposuctions that is fractionated and then cultured with a proliferative medium. This construct promotes ectopic bone tissue formation simplifying the limitations related to some inadequate properties of biomaterials used as scaffolds [47]. Until now, it is not possible to assess whether a technique is superior to another as homogeneity in the animal models is still required before applying it on a large scale in humans. However, little separates these techniques and many others from clinical application.

## 5. Conclusions

A significant amount of data from basic and translational research suggests that ADSC/SVF grafting and implantation is a promising strategy for bone regeneration. This complex tissue fraction holds ADSCs within an enriched trophic environment that sustains ADSCs biological properties and niche homeostasis. Current literature demonstrates that clinical application of SVF and ADSCs-derived grafts is safe and feasible though it cannot be routinely performed, as it requires experienced stem cells GMP manufacturing laboratories. Moreover, limitations imposed by national regulatory frameworks are severe in the case of non-homologous tissue implantation and non-homo-functionality application, thus affecting the number of the clinic studies on the topic. However, some clinical devices for minimal manipulation of the adipose tissue to isolate SVF and uncultured ADSCs currently represent a valid option to be exploited for the enhancement of long bones non-union healing. Further efforts are needed to investigate the role of ADSCs for the treatment of post-traumatic non-unions of the long bones.

## Figures and Tables

**Figure 1 ijms-23-03057-f001:**
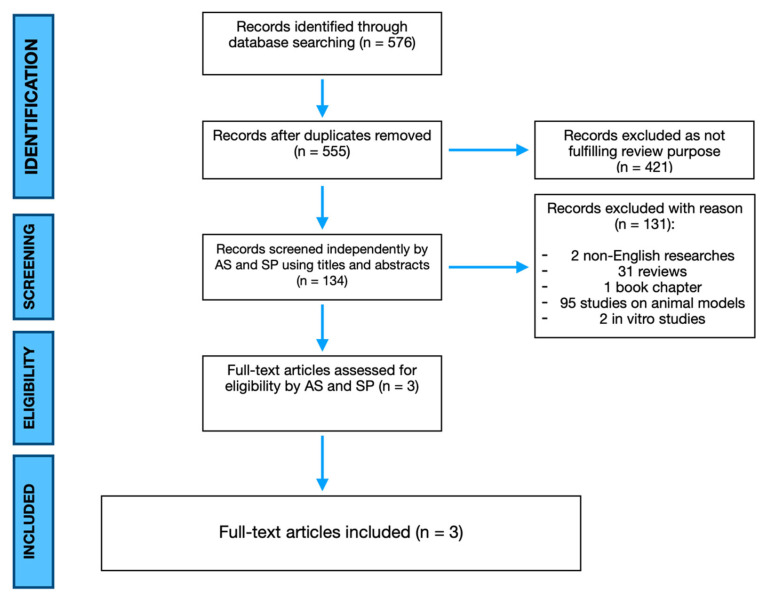
PRISMA flowchart highlighting search strategy and paper selection.

**Table 1 ijms-23-03057-t001:** Main demographical and clinical data from the included studies.

	N° of Patients	Age	Sex	Type of Lesion	Surgical Treatment	ADSCs Application	Outcomes **	Follow-Up	Complications
Khalpey Z. Et al. 2015	1	65	M	Sternal non-union with bone loss	Open reduction, augmentation and plate fixation	Injection of autologous SVF cells	Good	6 months	-
Veriter S. Et al. * 2015	11	18	8M, 3F	6 bone tumors, 4 congenital and 1 iatrogenic non-union	Wide oncological resection with growing prosthesis implantation or bone grafting	Scaffold-free osteogenic 3D adipose-derived graft	Good	28.7 months	Delayed wound healing Intercalary allograft infection Subcutaneous collection Cellulite
Dufrane D. Et al. * 2015	6	9.7	5M, 1F	3 bone tumors, 3 congenital pseudoarthrosis	Wide oncological resection with growing prosthesis implantation or bone grafting	Scaffold-free osteogenic 3D adipose-derived graft	Good	39 months	Intercalary allograft infection Screw and plate infection

* These Authors used the same cohort of study. ** No PROMs were used.

**Table 2 ijms-23-03057-t002:** Characteristics of the main bone grafts and substitutes in clinical application. Table published under permission of Bhatt R.A et al., 2012 and Greenwald A.S. et al., 2001 [37,38] through Copyright Clearance Center’s RightsLink® service.

		Osteo-Conduction	Osteo-Induction	Osteo-Genesis	Osteo-Integration	Structural Support	Disadvantages
**Autologous Bone Graft**	**Cancellous bone**	+++	+++	+++	+++	−	Limited availability, donor site morbidity, blood loss
	**Cortical Bone**	+	+	+	+	++++	Limited availability, donor site morbidity, blood loss
**Allogeneic Bone Graft**	**Cancellous bone**	+	+	−	++	−	Risk of disease transmission and rejection
	**Cortical Bone**	+	−	−	+	+++	Risk of disease transmission and rejection
	**Demineralized Bone Matrix**	+	++	−	++	−	Variable osteo-conduction
**Synthetic Bone Substitutes**	**Calcium solfate**	+	−	−	++	+	Rapid resorption, osteo-conduction only
	**Hydroxyapatite**	+	−	−	−	++	Slow resorption, osteo-conduction only
	**Calcium Phosphate Ceramic**	+	−	−	+	++	Osteo-conduction only
	**Calcium Phosphate Cement**	+	−	−	+	+	Osteo-conduction only
	**Bioactive Glass**	+	−	−	−		Bioactive osteo-conduction only
	**Poly (methyl-methacrylate)**	−	−	−	−	+++	Inert, exothermic, monomer-mediate toxic

## Data Availability

Not applicable.

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
