# Peer review of "Clinical Application of Adipose Derived Stem Cells for the Treatment of Aseptic Non-Unions: Current Stage and Future Perspectives—Systematic Review"

_ijms, 2022, doi:10.3390/ijms23063057_

Round 1
Reviewer 1 Report
The systematic review article is clear and straightforward and can be published with minor revision. However, it needs more information to be added.
- This article is a “systematic review” regarding the “Clinical application of adipose-derived stem cells for the treatment of aseptic non-unions: current stage and future perspectives”. However, nothing has been mentioned about the bone grafts available.
- Does the author have an idea about “the role of bone grafts concerning the healing of non-unions with ADSC”? Add a subtitle regarding this in the manuscript
- Include a table regarding the various types of bone grafts, with regards to source, properties, adverse reactions, types of techniques used for the manufacturing etc.,
- Most important are the advantages and disadvantages of different bone grafts.
- We know all the bone grafts are not suitable for “adipose-derived stem cells” in clinical use, add those details in the manuscript.
- Tabulate the commercial bone grafts suitable for the treatment of non-unions with ADSC.
- For the systematic review, as per the PRISMA flowchart paper selection, the author has selected 3 articles, is it enough to comprehensively convince the objectives?
Author Response
We are pleased to submit the revised version of the manuscript previously entitled “Clinical application of adipose derived stem cells for the treatment of aseptic non-unions: current stage and future perspectives. Systematic review.”
Reviewer’s #1 comments were highly appreciated. A detailed response structured as a point-by-point answer is therefore provided below.
Please note that all the altered passages are highlighted in using the track changes function in the revised manuscript.
We would be glad to consider any further suggestions and remarks.
Reviewer 1:
The systematic review article is clear and straightforward and can be published with minor revision.
Authors response:
We appreciated the kind comments and we would like to thank the Referee for the insightful consideration.
Points 1, 2, 4 and 5 are answered together because they all refer to the bone graft role in the treatment of non-unions.
- This article is a “systematic review” regarding the “Clinical application of adipose-derived stem cells for the treatment of aseptic non-unions: current stage and future perspectives”. However, nothing has been mentioned about the bone grafts available.
- Does the author have an idea about “the role of bone grafts concerning the healing of non-unions with ADSC”? Add a subtitle regarding this in the manuscript
- Include a table regarding the various types of bone grafts, with regards to source, properties, adverse reactions, types of techniques used for the manufacturing etc.,
- Most important are the advantages and disadvantages of different bone grafts.
- We know all the bone grafts are not suitable for “adipose-derived stem cells” in clinical use, add those details in the manuscript.
- Tabulate the commercial bone grafts suitable for the treatment of non-unions with ADSC.
Authors response:
As kindly suggested by the Reviewer we modified the discussion of the manuscript by adding a paragraph entitled “the role of bone grafts”. It is well known that the role of bone graft is pivotal for the non-union bone healing. We described the different available types of bone grafts and substitutes, highlighting some of the current applications. We added a table which summarizes some of the most important characteristics of the bone grafts and substitutes. We also mentioned some important application of engineering-based graft. However, we believe that a deeper discussion on bone grafts and substitutes, would be out of the scope of the present study.
- For the systematic review, as per the PRISMA flowchart paper selection, the author has selected 3 articles, is it enough to comprehensively convince the objectives?
Authors response:
We thank the Reviewer for this useful comment. Of course, we understand and agree that the low number of the collected studies represents a relevant limitation of the study (line 224). Nonetheless, this aspect could not be dominated by the investigators who can only collect, report, and interpret data. At present, the few reports published on the topic and the low number of the available clinical studies encourages further investigations on this issue.
Reviewer 2 Report
The review presents the current state of knowledge regarding the treatment of non-union fractures of bones with MSC cells derived from adipose tissue. The authors state that there is a lack of extensive research on this topic and discuss three reports that meet their search criteria. I think that it is not enough to cover 3 articles for a review, but indeed, very few reports have been published on this topic.
One comment: in Section 4.1 the authors described the work of Tay et al and stated that "only 72% of patients suffering non-union are able to return to their job after twelve months, in comparison to the 59% of those who have no other complications after surgical intervention ". Tay et al. wrote that 72% of patients with union had returned to work at one year compared to 59% of patients with delayed union or nonunion. The authors should correct this statement.
Author Response
We are pleased to submit the revised version of the manuscript previously entitled “Clinical application of adipose derived stem cells for the treatment of aseptic non-unions: current stage and future perspectives. Systematic review.”
Reviewer’s #2 comments were highly appreciated. A detailed response structured as a point-by-point answer is therefore provided below.
All the altered passages are highlighted in using the track changes function in the revised manuscript.
We would be glad to consider any further suggestions and remarks.
Reviewer 2:
The review presents the current state of knowledge regarding the treatment of non-union fractures of bones with MSC cells derived from adipose tissue. The authors state that there is a lack of extensive research on this topic and discuss three reports that meet their search criteria.
Authors response:
We thank the Reviewer for the time spent in reviewing our investigation and for the valuable suggestions provided to improve the overall quality of the review.
I think that it is not enough to cover 3 articles for a review, but indeed, very few reports have been published on this topic.
Authors response:
Dear reviewer, we are grateful to agree on the low number of the collected studies. As a matter of fact, this aspect is mentioned as a limitations of the study (line 224). However, be believe that the low number of the available clinical studies on the issue encourages further investigations on this open question.
One comment: in Section 4.1 the authors described the work of Tay et al and stated that "only 72% of patients suffering non-union are able to return to their job after twelve months, in comparison to the 59% of those who have no other complications after surgical intervention ". Tay et al. wrote that 72% of patients with union had returned to work at one year compared to 59% of patients with delayed union or nonunion. The authors should correct this statement.
Authors response:
We thank the Reviewer for this suggestion, and modified the paragraph highlighted in lines 177-180, accordingly.